# Partitioning Power Grid for the Design of the Zonal Energy Market While Preserving Control Area Constraints



**Marcin Blachnik** [1,*,†,‡] **, Karol Wawrzyniak** [2,‡] **and Marcin Jakubek** [2,*,‡]

1 Department of Industrial Informatics Akademicka 2A, Silesian University of Technology, 44-100 Gliwice, Poland

2 National Centre for Nuclear Research, Interdisciplinary Division for Energy Analyses, Andrzeja Sołtana 7, 05-400 Otwock-Świerk, Poland; karol.wawrzyniak@ncbj.gov.pl

* Correspondence: marcin.blachnik@polsl.pl (M.B.); m.jakubek@ncbj.gov.pl (M.J.); Tel.: +48-32-603-4270 (M.B.); +48-22-118-4412 (M.J.)

† Current address: Krasińskiego 8, 40-019 Katowice, Poland.

‡ These authors contributed equally to this work.

**Abstract:** The use of a zonal structure for energy markets across the globe is expanding; however the debate on how to effectively partition the grid into bidding zones is still open for discussion. One of the factors that needs to be addressed in the process of bidding zones' delimitation is the transmission system operators control areas. Merging parts of different control areas into one bidding zone can lead to multiple problems, ranging from political, through grid security concerns, to reserve control issues. To address it, this paper presents a novel grid partitioning method aimed at bidding zones delimitation that is based on clustering the power grid using an extended version of the standard agglomerative clustering. The proposed solution adds additional clustering rules when constructing the dendrogram in order to take into account the control areas. The algorithm is applied to the data which represents the locational marginal prices obtained from optimal power flow analysis.

**Keywords:** agglomerative hierarchical clustering; bidding zones delimitation; control areas; energy grid partitioning; locational marginal price; zonal energy market





## 1. Introduction

Currently, the zonal structure of the electricity market in Europe is undergoing a process of transformation [1]. Generally, a zonal market for electricity assumes that a given geographical region is divided into separate areas-bidding zones-across which the prices of electricity might differ, while within any bidding zone the price is constant. In the current organization of the European zonal market for electrical energy, the bidding zones follow closely— with some exceptions—the national borders. This "default" configuration of bidding zones is under dispute and is currently being reviewed [2], or even re-designed [3] by the organizations governing the European electricity market. If the current bidding zones configuration of the European market is to be changed, a reliable method to partition the energy grid that takes into account the needs of the zonal market organization, must be used.

The trading of energy on a zonal market is managed by an algorithm called Market Coupling (MC) [4,5], which governs the energy exchanges between the bidding zones, while the trade inside a bidding zone is assumed to be unconstrained (each bidding zone constitutes a copper plate per definition). The MC is intended to exploit the capacities of interconnections so that the energy flows optimally between bidding zones, i.e., ensure that the lowest-cost generators operate not only for local needs, but serve to increase the efficiency of the whole interconnected market. Still, how much efficiency could be increased by introducing a zonal market depends on how the bidding zones are delimited, and designing a robust bidding zone configuration constitutes one of the key issues in

adopting this form of energy market. (For a discussion on how the market outcomes depend on the bidding zones configurations see, for example, [6] or, in context of number of bidding zones, [7].) To this end, the approach used in the literature is based on machine learning methods-specifically, clustering-that is applied to some characteristics of the nodes or transmission lines of the grid.

In general, clustering is a group of methods aimed at identification of similar groups in the data [8,9]. The standard methods of clustering should however be modified in order to address the two specific prerequisites necessary to assure manageability of the bidding zones configuration by the zonal market.

The first, quite obvious prerequisite is that each bidding zone must constitute an electrically connected subset of the energy grid, so that the assumption about a bidding zone being a copper plate is plausible—we refer to this requirement as the topological constraints on clustering. This issue was already addressed in the context of zonal market division by [10], and most of the approaches to bidding zones delimitation discussed in Section 3 satisfy this requirement by design. The chosen approach to this requirement in the proposed methodology will be described in Section 4.

The second prerequisite is related to the fact that usually, the international power system in question is already divided either by country borders and/or by control areas managed by different transmission system operators (TSOs). (The technique presented in the paper can be easily adapted to take into account other geographical divisions, such as country borders or load-frequency control blocks, as long as each of those areas constitute a connected subset of the grid.) Neglecting the control areas' borders while delimiting the bidding zones might result in a division which is likely to include:

- bidding zones that are similar to control areas but include a subset of nodes from another control area,
- bidding zones encompassing parts of two or more control areas.

Such divisions can cause many issues of different natures: political (related to countries' energy policies), legal (related to contracts binding the energy producers, distributors and TSOs), and-last but not least-technical(related to security of the power system). (Although the assumption of each bidding zone being a copper plate is plausible from the market perspective, in reality the safety limits of the transmission lines inside a bidding zone must also be monitored and controlled by the corresponding TSO.) Attention has been brought to this prerequisite after a pioneering take on re-designing European bidding zones, conducted as the "First Edition of the Bidding Zone Review" by the European Network of Transmission System Operators for Electricity (ENTSO-E). The Final Report [3] from this endeavor mentioned that if the new bidding zones do not follow the borders of control areas (or balancing areas), the adaptations to the new bidding zones configuration may be more demanding ([3], pp. 83–84). Additionally, the Final Report cites that three TSOs (Polish PSE, Slovakian SEPS and Hungarian MAVIR) performed alignments to the proposed there bidding zone configuration justified by the notion that "the Single Control Area (CA) may contain more than one bidding zone and bidding zones may be built up from a few CAs but not parts of CAs" ([3], p. 181).

To the best of our best knowledge, the issue of taking into account control areas in the bidding zones' delimitation methodology has not been addressed so far in any publication related to partitioning of the electrical grid for the needs of zonal energy market.

In this paper, we propose a method that extends the work of Burstedde [10], which in addition to satisfying the topological constraints, also addresses the issue of partitioning the network grid into bidding zones in such a way that the control areas are taken into account-we refer to this requirement as the control area constraints on clustering.

The solution proposed in this paper uses hierarchical agglomerative clustering algorithm with built-in constraints on both the grid topology and the control areas. We show the validity of the proposed solution with an intuitive proof, and we present some clear numerical experiments to compare the bidding zones configurations obtained with and without the CA constraints.

Additionally, although the theoretical description of the clustering method in Sections 4 and 5 and the numerical examples given in Section 6 are presented from the perspective of clustering on a basis of a specific characteristic of the grid nodes (Location Marginal Price, LMP-described in more detail below), the proposed approach is not limited to this choice of clustering feature. Namely, the proposed approach can serve as a "meta-methodology" to bidding zones delineation, and can be applied to a wide range of nodes' (or, after adaptation, transmission lines') features or sets thereof.

The paper is organized as follows. In the next section we provide a definition of the control areas. The subsequent section provides a general review of clustering methods applicable to the problem of division of the energy grid into bidding zones, as well as the introduction of the metric space which is the input of a clustering algorithm. In Section 4 we briefly discuss standard agglomerative clustering and its modification addressing the topological constraints. Section 5 presents our extension to agglomerative clustering which supports the control area constraints, in the subsequent section we provide numerical experiments, and finally in Section 7 we summarize the proposed solution and the obtained results.

## 2. Control Area Constraints

As already introduced, the zonal energy market delimitation should answer the need to enable the electrical energy flows between bidding zones, and assure higher use of cheaper generators. However, an unconstrained delimitation can stand in conflict with the security policies of individual TSOs, so that the biding zones should not divide TSOs' control areas. Formally the control area is defined (as per [11], p. G-4) as "a coherent part of the UCTE interconnected system (usually coincident with the territory of a company, a country or a geographical area, physically demarcated by the position of points for measurement of the interchanged power and energy to the remaining interconnected network), operated by a single TSO, with physical loads and controllable generation units connected within the control area."

To cluster the energy market into bidding zones without violating the security aspects, three rules have been developed that must be met. These rules are (see Figure 1):

1.  it is allowed to split a given control area into smaller bidding zones (a single control area can be divided into any number of zones),
2.  it is allowed to combine two or more control areas into a bigger bidding zone (whole control areas can be merged into a bidding zone, or a bidding zone consisting of two or more whole control areas can be merged with another bidding zone consisting of one or more whole control areas),
3.  it is not allowed to combine parts of different control areas, or a part of a control area with another whole control area.

As a consequence, the results of the clustering algorithm that will be used during the delimitation must fulfill these rules.

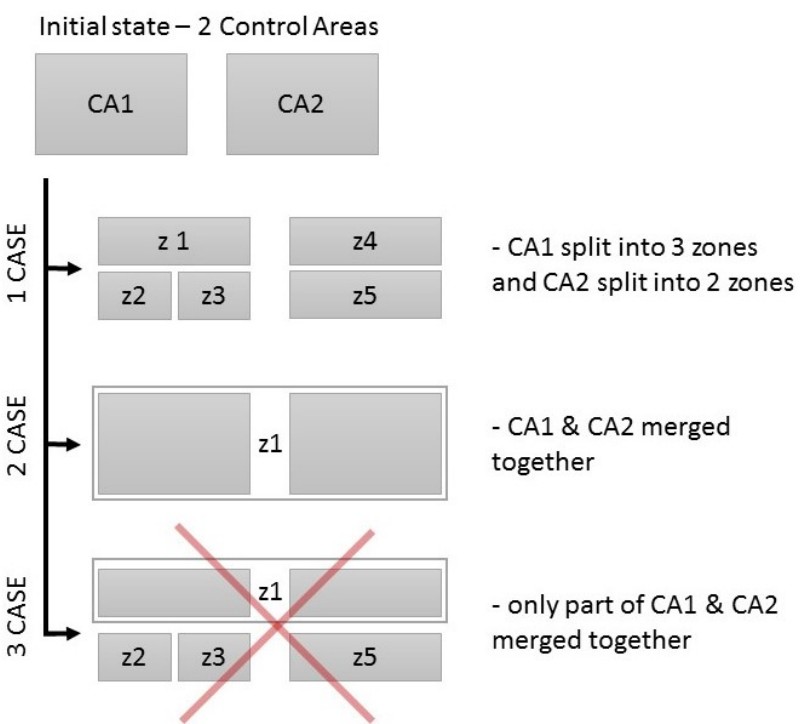

**Figure 1.** Allowed and forbidden bidding zones configurations with respect to control area constraints.

## 3. Energy Grid Clustering Methods

The problem of energy grid partitioning can be divided into two sub problems. The first one is the selection of the proper algorithm which will be used for the input data decomposition, and the second one is related to determining the feature space which will be used as an input to the clustering algorithm. These two sub problems will be discussed in the following subsections.

### 3.1. Grid Partitioning Methods

In general, partitioning of the interconnected energy grid can be perceived as a problem from the general family of graph or network clustering, for which many methods have been developed so far to address various contexts of the partitioning, also other than designing a zonal energy market. A nice overview can be found in [12] where the authors divide the methods into four groups: hierarchical clustering, partitioning clustering, spectral clustering and divisive algorithms. Examples of applications for power grid partitioning can be found in [13], where in chapter 6 network decomposition methods based on community structure analysis are presented. Other attempts at network clustering can be found in papers discussing vulnerability analysis and assessment such as [14] where based on scale free network properties, the authors provide decomposition of the European network. Many attempts at grid partitioning can be also found for the purpose of intentional islanding to limit cascade power failures, an example can be found in [15], where the authors used spectral methods to separate poorly connected parts of the network. Namely, the method first partitions network grid based on classical methods, and then applies simulating annealing Monte Carlo algorithm to optimize resulting clusters for internal connectivity and power self-sufficiency as well as possible other properties. (Although the method proposed by [15] is not designed particularly for zonal market design problems, it seems it can be adapted to this challenge with limited effort).

Out of large group of network clustering method, one of the most commonly used are hierarchical clustering methods, which has several advantages. It allows clustering input data which can be described in any feature space, where the structure of the network is used to define the neighbourhood in terms of nodes. It also allows visualizing the structure

of the data by the dendrogram. Finally, it does not require to determine the number of clusters beforehand, this value can be selected in a post-processing step.

### 3.2. The Feature Space of the Zonal Market

Still, the methods mentioned above were designed to perform the partitioning from a different perspective than the needs of the zonal market. For example they identify weakly connected nodes or subnets, but to be applicable to energy market design they need to take into account economic- or market-based characteristic of the system.

An interesting early approach to bidding zone delimitation is the idea of using the maximum spread of nodal prices within groups using the tabu search heuristic proposed by [16], which addresses the specific needs of the British Electricity Trading and Transmission Arrangements (BETTA). However, the current arrangement chosen for, among others, the integrated market of EU countries takes on a different approach based on the Market Coupling algorithm, which calls for yet another approach.

We thus focus on a bidding zone design for a *Market Coupling*-based energy market. Classical graph partitioning methods as well as complex network analysis methods ignore Kirhoff's laws and other physical constraints such as lines' thermal limits. They also completely neglect the market layer in the clustering process. To design a delimitation of bidding zones which would be most easily manageable by the MC algorithm, a method of delimiting bidding zones should take into account both the physical flows over the transmission lines and their capacity limits, as well as the spatial cost structure of the energy generation in relation to the energy demand distribution. To address these factors, two main methodologies were designed. (Other approaches apart from those general two include, for example, a multilevel optimization problem for obtaining zone design which aims at welfare optimum—see [17], or a clustering technique that focuses on decreasing the costs of redispatching needed in the existing zonal partition—see [18].)

The first one is based on the Locational Marginal Prices (LMP-based) [10,19], and takes directly into account the (nodal) market layer, the realizations of which include the physical layer in an indirect way. An LMP, or nodal price, represents the local value of energy in a given place in the network, i.e., how the overall delivery cost will change after supplying an extra 1 MW of power to this particular node-physical point in the transmission system, where energy can be injected by generators or withdrawn by loads. This price consists of the cost of producing the energy used at this node and the cost of possible congestions arising from delivering it, which may affect the overall system costs. Therefore, LMPs are grid nodes' features which allows us to separate locations into higher and lower price areas if congestion occurs between them. The LMP-based approach to bidding zones delimitation generally works in the following way. First, the LMPs are calculated by the Optimal Power Flow (OPF) algorithm, in which they are represented by the Lagrange multipliers on real power mismatch in the nodes. Next, on the nodes' characteristics representing LMPs a hierarchical clustering algorithm based on Ward's minimum variance is used to aggregate nodes of similar prices into biding zones with additional constraints applied to represent the topological structure of the energy gird [20]. As a result, clustering should assign nodes that span congested lines into different clusters, because congestion differentiates the LMP values across the grid. Some more complex LMP-based techniques include, for example, weighting the nodes with respect to their relevance in energy infeed or demand (see [21]), or providing an insight into the optimal number of bidding zones (see [22]).

The second approach is based on Power Transfer Distribution Factors (PTDF-based), which directly take into account the physical flows over the network structure which are a result of market operations. A detailed description of the PTDF methodology can be found in [23,24]. A comparison of various clustering methods applied to both approaches can be found in [25].

Both the LMP- and PTDF-based methodologies take into account the topological constraints of the network but does not currently address the control area constraints. Below we provide a way in which the control area constraints can be added to the LMP-

based bidding zones delimitation, although the PTDF-based method can be extended in an analogous manner as well.

To present formally the clustering method, in what follows we assume that the data of the energy grid are represented as an *n*-element dataset $\mathbf{X} = [\mathbf{x}_1, \mathbf{x}_2, \ldots \mathbf{x}_n]$ where each element is an *m* dimensional vector $\mathbf{x}_i = (x_i^1, \ldots, x_i^m) \in K \subset \Re^m$ of the object's *m* numerical features. We assume that the feature space *K* is metric, and we denote the distance function as $\rho(\mathbf{x}_i, \mathbf{x}_j)$. In the application to energy network, the objects are the nodes (buses), with their features space being one- or multi-dimensional LMP space, or multi-dimensional PTDF space, and $\rho$ the Euclidean distance on $\Re^m$. (A multi-dimensional LMP space can be obtained by collecting LMPs values for different grid conditions ex. peak/off-peak conditions, winter/summer demand patterns etc.)

In the next section, we will provide a brief description including the topological constraints into agglomerative hierarchical clustering, which will be further extended to the control area constraints in the following section.

## 4. Agglomerative Hierarchical Clustering

Agglomerative hierarchical clustering, which will be discussed here was developed in the 1960s by J. H. Ward [20]. Its name is based on two characteristics of how the method works. Firstly, it starts by representing each object $\mathbf{x}_i$ as a single-element cluster, and then recursively merges (agglomerates) the two closest clusters into one, up to the point when a single cluster encompassing all of the objects is achieved. Secondly, in the process the method builds a tree hierarchy of clusters' mergers, where each node of the tree has exactly two child nodes (a binary tree).

The basic algorithm is widely known and is explained in almost every textbook on machine learning or clustering (cf., for example, [8,9,26]), therefore we will present it only to the extent that is necessary to highlight the differences between the standard method and our modified algorithm. For those who are interested in implementation details we refer to [27].

The sketch of the algorithm is provided in Algorithm 1, where:

- **D** is the matrix of inter-cluster distances; at the beginning it is the distance matrix between all of the objects, $\mathbf{D}_{i,j} = \rho(\mathbf{x}_i, \mathbf{x}_j)$, in later steps the distances are derived for the objects the clusters consist of using some linkage function (described below), namely $\mathbf{D}_{i,j} = linkage(C_i, C_j)$,
- GETCLOSESTCLUSTERS() is the function which determines the closest pair of clusters (denoted $C_a$ and $C_b$) to link into a single cluster in the current step, with their distance denoted by $d_{a,b} = linkage(C_a, C_b)$,
- *tree* is a table representation of a binary tree with the history of cluster mergers. A *k*-th row of the table consists of two cluster id's ($C_a$ and $C_b$) which are being grouped at the step *k* and the distance between these linked clusters, $d_{a,b}$.
- UPDATECLUSTERSDISTANCE() is the function which recalculates the distances between clusters after the merging of $C_a$ with $C_b$-as there will be one cluster less after merging, the updated distance matrix $\mathbf{D}^*$ will be of one dimension less than the matrix **D**.The update of the clusters' distances is usually done using Lance-Williams algorithm [28], which speeds up the calculations.

After performing $n - 1$ steps, where *n* is the number of objects, we obtain a tree which on each level (row) keeps track of the two nearest clusters being merged at a given step. The tree can be simply visualized using a dendrogram plot as presented in Figure 2, where the height of tree branches represents the distance between the clusters being merged. In the standard agglomerative clustering the tree is usually monotonic (the distance between merged clusters increases along the steps of the procedure), which is represented on a dendrogram by the branches not crossing each-other (a link between two merged clusters is placed higher than the links which represent previous mergers). Interestingly, after incorporating the topological constraints or the control area constraints this property might not hold anymore, which will be discussed later.

---

**Algorithm 1** Agglomerative clustering algorithm

---

    **function** BUILDTREE( *tree*, **D**)
        **if** $\|tree\| == n - 1$ **then**
            **return** *tree*
        **end if**
        $\{C_a, C_b, d_{a,b}\} =$ GETCLOSESTCLUSTERS(**D**)
        $tree = tree \cup [C_a, C_b, d_{a,b}]$
        $\mathbf{D}^* =$ UPDATEDISTANCEMATRIX($\mathbf{D}, C_a, C_b$)
        $tree =$ BUILDTREE($tree, \mathbf{D}^*$)
        **return** *tree*
    **end function**

---

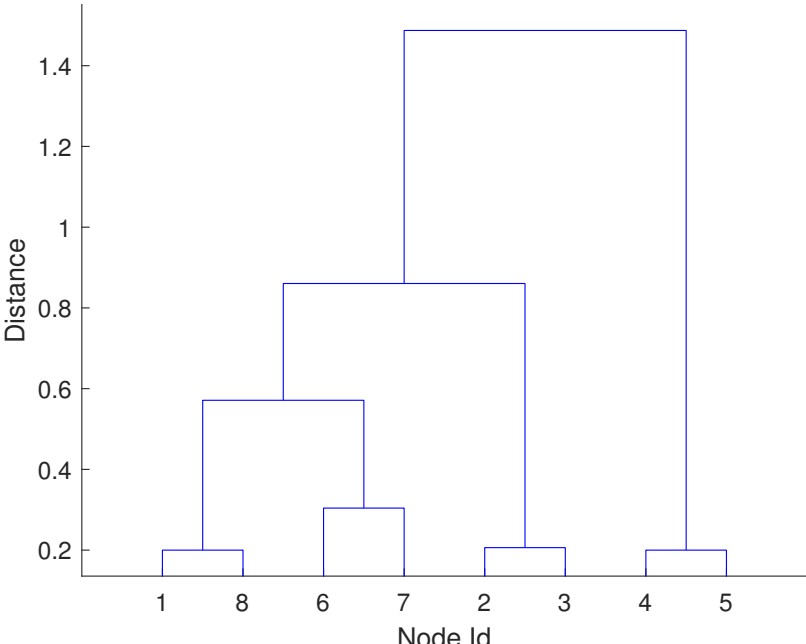

**Figure 2.** Example dendrogram tree plot.

    The key element of agglomerative clustering is the choice of the linkage function, namely the function which translates (or aggregates) the distances between individual objects belonging to two clusters into the distance measure between these clusters. A broad set of linkage functions was developed, each preserving different properties of the distances between objects. The most popular linkage functions, presented in Table 1, are: *single linkage* based on the smallest distance between every pair of objects in the two clusters, *complete linkage* based on a minimum of the two furthest objects in the two clusters, *average linkage* (perhaps the most intuitive) based on the average distance between every pair of objects in the two clusters, *centroid linkage* in which each cluster is represented by its *centre* (prototype or centroid) and the clusters' distance is measured as the distance between their centres.

**Table 1.** Comparison of the most popular linkage functions.

| Linkage Function | Name |
|---|---|
| $d(C_a, C_b) = \displaystyle\min_{\mathbf{x}_i \in C_a, \mathbf{x}_j \in C_b} \left( \rho(\mathbf{x}_i, \mathbf{x}_j) \right)$ | single linkage |
| $d(C_a, C_b) = \displaystyle\max_{\mathbf{x}_i \in C_a, \mathbf{x}_j \in C_b} \left( \rho(\mathbf{x}_i, \mathbf{x}_j) \right)$ | complete linkage |
| $d(C_a, C_b) = \displaystyle\mathop{\text{mean}}_{\mathbf{x}_i \in C_a, \mathbf{x}_j \in C_b} \left( \rho(\mathbf{x}_i, \mathbf{x}_j) \right)$ | average linkage |
| $d(C_a, C_b) = \rho\left( \mathbf{x}_{C_a}, \mathbf{x}_{C_b} \right)$ | centroid linkage, where: $\mathbf{x}_{C_a} = \frac{1}{\|C_a\|} \sum_{\mathbf{x}_i \in C_a} \mathbf{x}_i$ centroid of cluster $C_a$ $\mathbf{x}_{C_b} = \frac{1}{\|C_b\|} \sum_{\mathbf{x}_j \in C_b} \mathbf{x}_j$ centroid of cluster $C_b$ |
| $d(C_a, C_b) = \frac{\|C_a\| \cdot \|C_b\|}{\|C_a\| + \|C_b\|} \rho\left( \mathbf{x}_{C_a}, \mathbf{x}_{C_b} \right)$ | Ward linkage |

Finally, the Ward linkage function is similar to the centroid linkage but with an extra normalization factor. As discussed in [29], using the Ward linkage results in clusters which are more similar in size, while for other linkage functions it often happens that the cardinalities of the final clusters varies such that very large clusters appear next to very small ones.

Agglomerative hierarchical clustering has one disadvantage which is high computational complexity reaching $O(n^3)$ and high space complexity ($O(n^2)$). So far many methods have been developed to boost these limits, for example: the previously mentioned distance update formula [28], more efficient nearest-neighbour search [30] reaching the $O(n^2 \log(n))$ complexity, or applications of so-called local sensitive hashing [31] for large scale single linkage hierarchical clustering [32]. Still, the computational and / or space complexity of those variants of agglomerative clustering is not comparable with many of the so-called model-based clustering methods, which can have linear time and space complexity [33]. Apart from these drawbacks, hierarchical clustering possess many important advantages over model-based clustering algorithms:

- it doesn't require a predetermined number of clusters as an input of the algorithm-this number can be derived afterwards, moreover,
- it provides a nice visualization of the structure and relations between the clusters' distances, and plotting the dendrogram allows us to determine the plausible number of clusters post-ante,
- by changing the linkage function it allows us to influence the shape and properties of the resulting clusters.

An additional advantage of agglomerative hierarchical clustering -and the reason this clustering method was chosen in the approach to bidding zones delimitation presented in this paper- is the possibility to incorporate external constraints into the merging process [34], among which the topology constraints are crucial in designing the energy bidding zones, and will be briefly discussed now.

### 4.1. Preserving Topology Constraints

We assume that apart from the objects' (grid nodes') features space, the input data includes a graph which represents the topology of the energy network. Namely, let $G = <\mathbf{X}, E>$ be an undirected connected graph with nodes being the objects from set $\mathbf{X}$ and where $E$ are the edges representing the transmission lines between the nodes in the grid.

We include the topological constraints into the agglomerative hierarchical clustering in a manner akin to [10]. Specifically, we modify the linkage function so that if there does not exist a direct link between clusters $C_a$ and $C_b$ (no pair of nodes from each of the clusters is connected by an edge of the graph $G$), the distance between $C_a$ and $C_b$ is equal to infinity. With the distances between unconnected clusters being converted to infinity,

such clusters would not be merged (at least under the assumption that the whole graph is a connected one).

Namely, instead of distance $d(C_a, C_b)$ we use

$$d'(C_a, C_b) = \begin{cases} d(C_a, C_b) & \text{if } \exists E(\mathbf{x}_i, \mathbf{x}_j) \text{ for } \mathbf{x}_i \in C_a, \mathbf{x}_j \in C_b \\ +\infty & \text{otherwise.} \end{cases} \tag{1}$$

A straight extension of topological constraints to standard algorithm (Algorithm 1), presented in Algorithm 2, requires only (i) a modification of the cluster distance function from $d(.,.)$ to $d'(C_a, C_b)$ in the UpdateConnectedClustersDistance function, which is analogous to UpdateClustersDistance in Algorithm 1, and (ii) passing a data structure holding the representation of the edges of the graph, *E*. (A very efficient implementation of this algorithm which we used in our experiments can be found in Python scikit-learn package: http://scikit-learn.org/stable/modules/clustering.html#hierarchical-clustering, accessed on 11 February 2021).

---

**Algorithm 2** Agglomerative clustering algorithm with topology constraints

---

　**function** BUILDTREETOP( *tree*, **D**, *E*)

　　**if** $\|tree\| == n - 1$ **then**

　　　**return** *tree*

　　**end if**

　　$\{C_a, C_b, d_{a,b}\}$ =GETCLOSESTCLUSTERS(**D**)

　　$tree = tree \cup [C_a, C_b, d_{a,b}]$

　　$\mathbf{D}^* = $ UPDATECONNECTEDCLUSTERSDISTANCE($\mathbf{D}, C_a, C_b, E$)

　　$tree = $ BUILDTREETOP(*tree*, $\mathbf{D}^*$,E)

　　**return** *tree*

　**end function**

---

It is worth noting that including topology constraints has a positive influence on the algorithm by radically reducing computational complexity-instead of calculating all inter-cluster distances, only distances between connected clusters need to be calculated. This issue is especially important in clustering very large energy grids, such as the pan-European network.

Interestingly, including topology constraints can cause non-monotonicity of the dendrogram tree. Namely, it may happen that at a later stage of the clustering process two clusters are merged which have a lower distance between them than the clusters merged previously.

For an example, see Figure 3, where an example input data is shown together with the graph topology, and two dendrograms. Dendrogram (b) was obtained without any constraints during clustering, while the second dendrogram (c) was obtained during clustering with the constraints representing the graph topology. The red line on dendrogram (c) indicates the non-monotonicity.

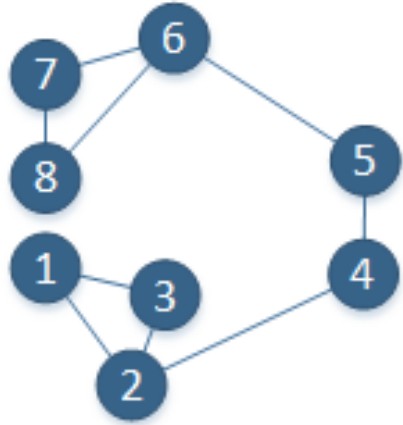

(**a**) Example data

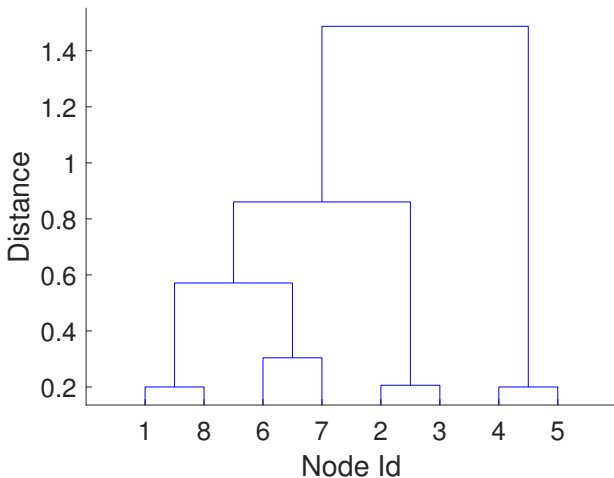

(**b**) Monotonic

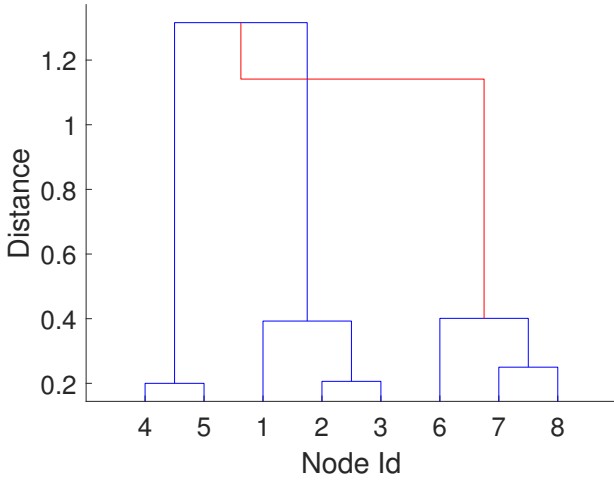

(**c**) Non monotonic

**Figure 3.** Example showing the effect of constraints influencing monotonicity of the dendrogram. (**a**) Example data with links defining the constraints; (**b**) dendrogram obtained without any constraints on graph topology; (**c**) dendrogram obtained when clustering preserving graph structure.

Interpretation of this phenomenon is rather simple. Consider two nodes of the network which lie next to each other in the feature space (nodes $N1$ and $N8$ in Figure 3), but do not have a direct link (line) connecting them. The only path connecting nodes $N1$ and $N8$ goes through nodes $N4$ and $N5$, so firstly these two nodes have to be connected to one of the clusters (on the dendrogram they are first connected to cluster $\{N1, N2, N3\}$), which allows us to merge cluster $\{N1, N2, N3, N4, N5\}$ with the cluster $\{N6, N7, N8\}$ in the next step. Considering the Ward linkage function, this can be interpreted as follows: merging cluster $\{N4, N5\}$ with $\{N1, N2, N3\}$ increases inter cluster variance (as nodes $\{N4, N5\}$ lie far from the rest), while finally joining cluster $\{N6, N7, N8\}$ to the already connected nodes decreases the total variance of the new cluster. In general, the lack of monotonicity of the dendrogram is quite common and, for example, it also appears when centroid linkage is applied even without any topological constraints. In the case of Ward linkage without constraints the algorithm is sure to generate monotonic dendrograms, but constraints may result in a lack of monotonicity. This lack of monotonicity indicates that usually one of the clusters before the merge has a relatively large variance caused by the extreme features' values of the nodes (in our case, large spread of LMPs) inside it like in the example above, and after the merge the distribution of a new cluster is less affected by the extreme values.

Summing up, the problem of lack of monotonic growth of the dendrogram may provide an important insight into the interpretation of relations between groups of buses and their connectivity as described above, but it does not question the validity of hierarchical clustering methodology. Here the hierarchy of the tree remains unchanged, because formally the algorithm returns a binary tree, and only the interpretation of the dendrogram requires more careful analysis.

## 5. Applying Control Area Constraints

As before, we represent the energy grid as a graph $G = <\mathbf{X}, E>$. We assume that a control area, denoted $G(CA_i)$ where $i \in \{1, 2, \ldots, c\}$ and $c$ is the number of control areas, is a connected subgraph of $G$, and that the set of all control areas-treated as subsets of $\mathbf{X}$-constitute a disjoint cover of $\mathbf{X}$. In real life the assumption on control area being a connected subset of $G$ may not hold, but it is very rare. In Europe there exists one control area (Amprion (RWE) in Germany) which is composed of two regions which are not connected by a direct transmission line. For the needs of the study, such control area needs to be treated as two separate control areas, each constituting a connected subgraph of the grid.

It is now important to analyze the relation between the three control area constraints (cf. Section 2) and the dendrogram tree (denoted as *tree*) produced by clustering of the energy grid. The third constraint states that it is not allowed to combine parts of different control areas-from that and from the assumption on control areas being disjoint sets, we induce that each control area should be represented as a single subtree, denoted $treeCA_i$, of the dendrogram *tree*, which can be denoted as ($treeCA_i \in tree$) and that the subtrees $treeCA_i$ do not overlap $\underset{i,j;i \neq j}{\forall} treeCA_i \cap treeCA_j = \varnothing$.

Control area constraints number 1 and 2 are also supported by the dendrogram tree if each control area is represented by single dendrogram tree $treeCA_i$. When cutting (flattening) the tree to determine clusters from the dendrogram, each subtree $treeS_i$ fulfills one of the two following cases:

**Case 1** If a subtree $treeS_j$ is a part of $treeCA_i$ ($treeS_j \in treeCA_i$), then we obtain another subtree $treeS_k$ which would complement $treeCA_i$, such that $treeS_k = treeCA_i \setminus treeS_j$. This fulfills condition 1 of control area constraints, namely a single control being divided into smaller parts.

**Case 2** If subtree $treeS_j$ includes $treeCA_i$ ($treeCA_i \in treeS_j$), then, from the fact that each control area is represented by a single tree, we obtain that the remaining tree $treeS_k = treeS_j \setminus treeCA_i$ (complement of $treeCA_i$) must include one or more other control areas $treeCA_k \in treeS_k$, namely we fulfill condition 2 of control area constraints.

Above, we showed that the algorithm satisfying control area constraints needs to have a separate tree for each *CA*, thus we build a separate dendrogram tree for each control area, and then combine them, following the tree growing process. This can be expressed as in Algorithm 3.

---

**Algorithm 3** Agglomerative clustering algorithm with control area constraints

---

    **function** BUILDTREECA( $\mathbf{X}, E, CAs$)

        $tree = \varnothing$

        **for** $i = 1$ to $c$ **do**

            $\mathbf{D}^* =$ DISTANCEMATRIX($\mathbf{X}(CA_i)$)

            $E^* = E(CA_i)$

            $treeCA_i =$ BUILDTREETOP($\varnothing, \mathbf{D}^*, E^*$)

            $tree = tree \cup treeCA_i$

        **end for**

        $\mathbf{D}^{\#} =$ DISTANCEMATRIX($\mathbf{X}, CA$)

        $tree =$ BUILDTREETOP($tree, \mathbf{D}^{\#}, E$)

        $tree =$ REORDERTREENODES($tree$)

        **return** $tree$

    **end function**

---

Here we assume that the input of the *buildTreeCA* method is the input space $\mathbf{X}$ (for example, the LMPs characterization of grid nodes); $E$ is a representation of the edges in graph $G$ (for example, in a form of an adjacency matrix), and the last argument *CAs* is the collection of sets of objects (nodes) associated with each of the $c$ control areas (for example, in the form of $c$ lists with numbers of nodes that belong to each of the control areas). At the beginning, the tree structure is empty. The algorithm starts by iterating over control areas, and for each control area $CA_i$ it builds a distance matrix $\mathbf{D}^*$ for nodes within this control area (using the distance with topology constraints, $d'(.,.)$), and receives a subset of the connections $E$ which represents links within given $CA$, $E^*$. These arguments are used as an in input to the *buildTreeTop* method described above in Algorithm 2. This method returns a dendrogram tree $treeCA_i$ which will be a subtree of the final dendrogram tree, so all subtrees $treeCA_i$ are collected together within the *tree* variable. After the loop the algorithm once again calculates the distance matrix $\mathbf{D}^{\#}$ but this time it represents distances between control areas each treated as a cluster. Next, the function *buildTreeTop* is called on again to merge subsequently the most similar pairs of CAs into clusters. The very last step of the algorithm reorders the nodes of the tree.

This procedure does not change the shape of the tree but changes the enumeration of tree nodes. This step is important from the tree cropping perspective as the algorithm starts from the root tree node and recursively evaluates tree nodes in the order they were created. In our algorithm this order is violated because the final tree which "glues" all the control area subtrees creates tree nodes after the subtrees were created. In other words, considering the dendrogram presented in Figure 4, the links (tree nodes) plotted black would be created after the top most green tree nodes. After the reordering routine, the lower black link is pushed down such that the top-most green links would appear higher in the tree, and the tree cropping algorithm would be able to split the green control area into two bidding zones before disconnection of the remaining control areas. An implementation of this algorithm is shown in Algorithm 4 where a single node of the tree is represented by a triple $node = <a, b, d_{a,b}>$ where $a$ and $b$ are references to the child tree nodes being merged and $d_{a,b}$ describes the distance between these tree nodes, and the final tree is represented by a list of tree nodes $[node_1, node_2, \dots, node_{n-1}]$.

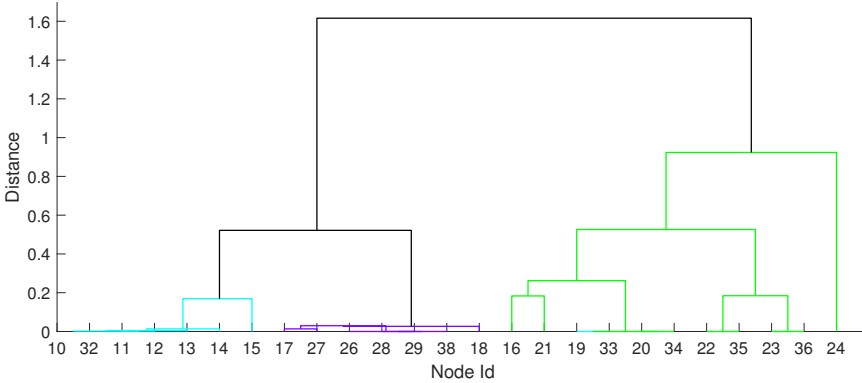

**Figure 4.** An example dendrogram where colours represent subtrees obtained for three control areas and the top part of the tree (marked black) merges all subtrees. Note that the bottom links of the black subtree start earlier than the top links of the green subtree.

---

**Algorithm 4** Pseudo-code of the function used to reorder nodes of the tree

---

> **function** REORDERTREENODES( *tree*)
>> *sortedTree* =SORTACCORDINGTODISTANCE(*tree*)
>> *newTree* = ∅
>> **for** *node* : *tree.leafNodes* **do**
>>> *newTree* = *newTree* ∪ *node*
>>> *sortedTree* = *newTree* \ *node*
>>
>> **end for**
>> *i* = 1
>> **while** !*isempty*(*sortedTree*) **do**
>>> *node* = *sortedTree*(*i*)
>>> **if** *newTree*.CONTAINS(*node.a*) &
>>>> *newTree*.CONTAINS(*node.b*) **then**
>>>> *sortedTree* = *sortedTree* \ *node*
>>>> *newTree* = *newTree* ∪ *node*
>>>> *i* = 0
>>>
>>> **end if**
>>> *i* = *i* + 1
>>
>> **end while**
>> **return** *newTree*
>
> **end function**

---

In this algorithm, we first sort nodes of the tree according to the distance $d_{a,b}$, then a new empty tree is created, and all the leaf nodes are added to the *newTree* structure as well as removed from the *sortedTree*. Then starts the main loop, in which we read *i*-th tree node from *sortedTree* and check if its child tree nodes already appear in the *newTree*. If the condition is valid, we remove this tree node from the *sortedTree* and we add it to the *newTree*. If the condition is not valid, we take the following tree node. The algorithm ends when *sortedTree* does not contain any nodes.

Compared to the standard agglomerative hierarchical clustering (cf. Section 4), the agglomerative clustering algorithm with control area constraints described above has lower computational and space complexity, as the subset of objects that can be connected in each step is greatly reduced: to nodes/clusters that are parts of one control area only, or to clusters representing whole control areas (or their aggregation). Thus, the complexity

depends not on the overall number of nodes in the grid ($n$), but on ($n^* + c$), which is the highest number of nodes in a control areas, denoted $n^* = max\{n_1, \ldots, n_c\}$, where $n_i$ is the number of nodes in a control area $i$, plus the number of control areas, $c$. This allows the application of this algorithm to fairly large networks.

## 6. Numerical Experiments

To show properties of the described algorithm we performed a series of experiments on a well known *IEEE39* case for which we manually defined control areas. This case consists of 39 nodes (among them 10 are generators) and 46 are branches (Figure 5).

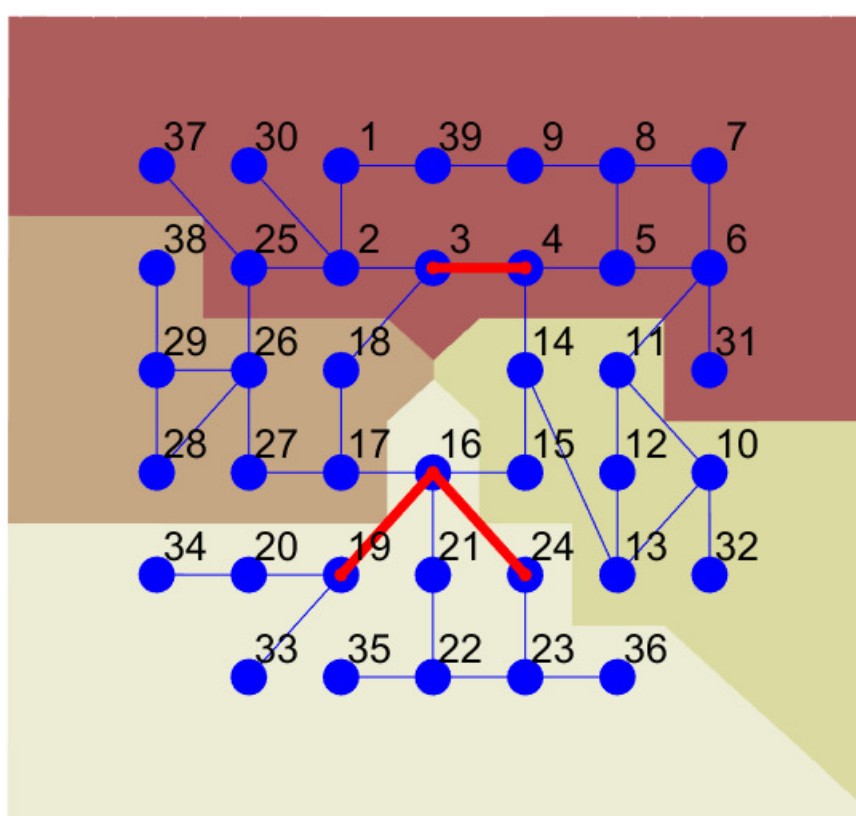

**Figure 5.** IEEE Case 39 with congested lines marked in red.

We introduced four control areas, each marked with a different background colour. Since one of the determinants of the zonal market (as described in Section 1) is the separation of areas influenced by congestions, we manually overloaded some of the lines. This was achieved by reducing line limits or modifying the consumption of active power in some of the nodes compared to the original *IEEE39* case. All dataset containing MatPower solved cases, with all modifications are available on http://prules.org/materials/ca/ accessed on 11 February 2021. Then, the *MatPower* [35] software was used to perform optimal power flow analysis resulting in LMPs values (as was mentioned above, also other bus descriptors, e.g. based on PTDFs can be used to form the clustering feature space). LMPs were then clustered using two approaches, one without control area constraints and the second one preserving control area constraints. Obtained dendrograms are shown in Figure 6, where to increase comprehensibility subtrees obtained for each control area are coloured with basic colours such as green, red, cyan and magenta and background colours which correspond to the colours used in Figure 5 to indicate control areas, while the final subtree of the dendrogram is plotted black (see Figure 6b). For dendrogram obtained without CA constrains, the entire dendrogram is blue.

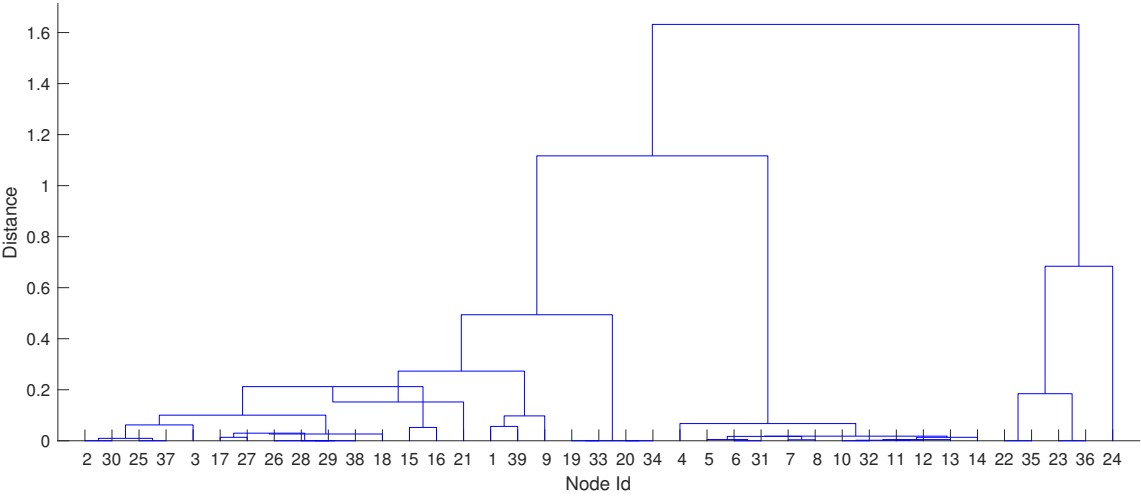

(**a**) Without control area constraints

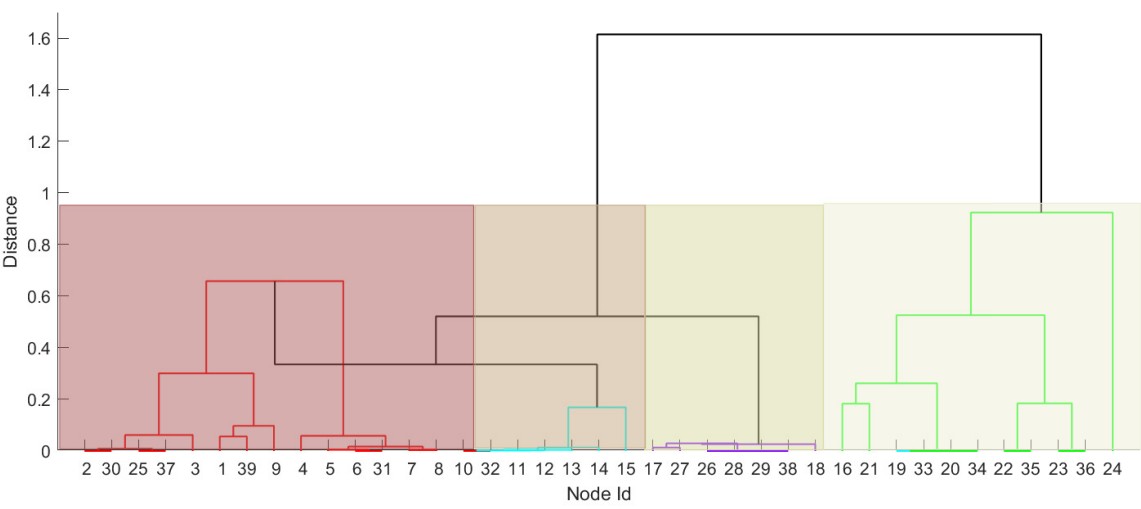

(**b**) With control area constraints

**Figure 6.** Comparison of the dendrograms obtained without (**a**) and with (**b**) control area constraints. Colours in figure (**b**) represent dendrograms obtained for each control area.

The bidding zones delimitations are visualized in the two-column Figure 7: the left column shows the output of the algorithm which did not preserve control area constraints, and the right column with. The rows represent divisions into 2 (top), 3 (middle) and 6 (bottom) bidding zones, with the bidding zones being marked by different node colours.

Two clusters

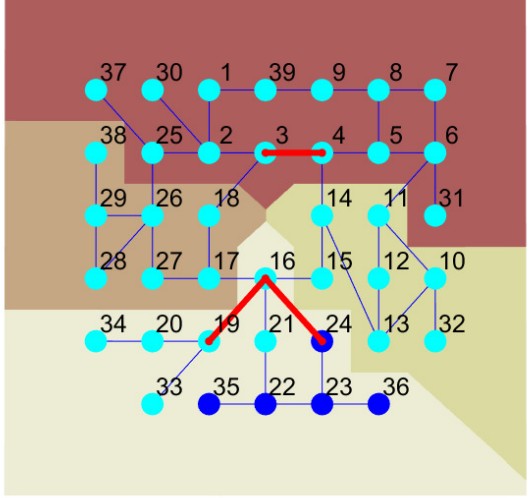 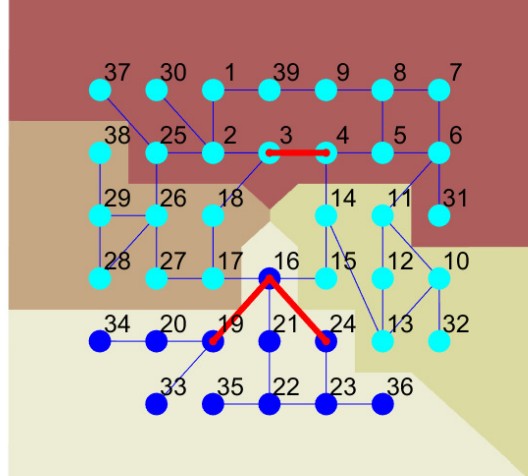

Three clusters

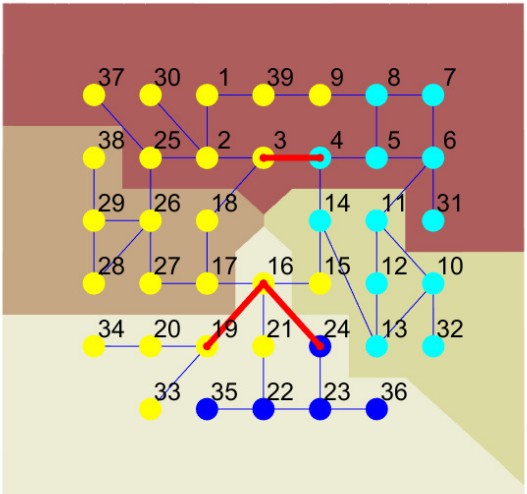 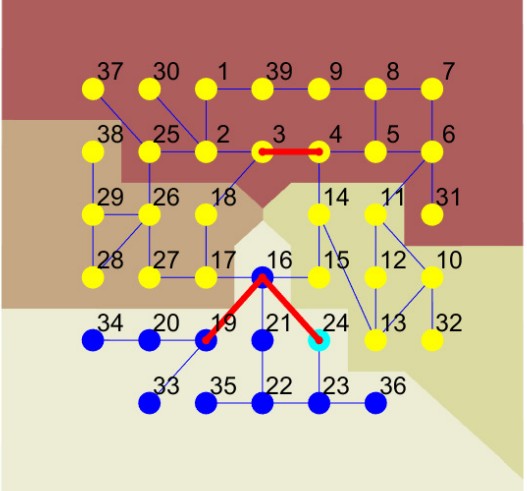

Six clusters

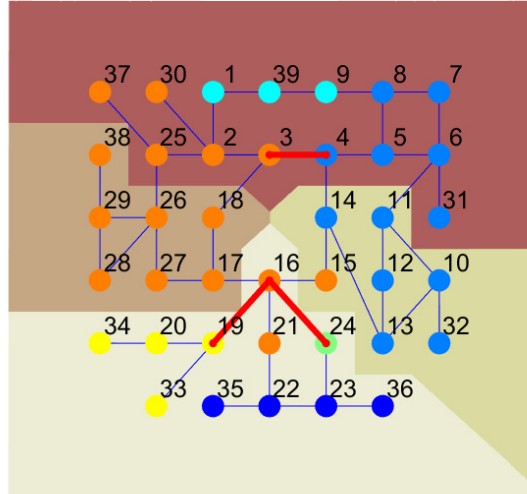 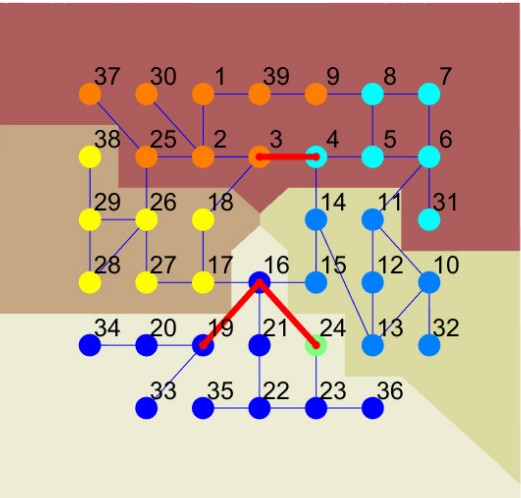

(**a**) Without control area constraints　　　　(**b**) With control area constraints

**Figure 7.** Comparison of bidding zones delimitation without (**a**) and with (**b**) control area constraints for 2 (**top figures**), 3 (**middle figures**) and 6 (**bottom figure**) clusters. In the plot background colour represents control areas and node colour indicates obtained bidding zones delimitation.

The CA at the bottom, which subtree is marked in green, constitutes a good example of differences between the two approaches. It is characterized by the largest value of the linkage metric (the green sub-tree is the highest), reflecting relatively large variety among LMP prices inside this CA. Compared to the dendrogram without constraints (a), one can see that an analogous sub-tree only exists partially. It indicates that some of the nodes from green CA are closer in terms of the linkage metric to nodes in other CAs than to each other. However, the third control area constraint does not allow the connecting of parts of different control areas into a bidding zone. This forces the algorithm to group all the nodes from this CA together before merging with other CAs is possible. Due to the high heterogeneity of LMPs in the bottom CA and much stronger similarities between nodes in other CAs, the nodes are merged with other clusters in the very latest iteration of the algorithm which controls the CA constraints. It means that if we require a division into two bidding zones, then one bidding zone is constituted by green CA (the bottom one in Figure 5) and another bidding zone by all other CAs grouped together. This is shown in the first row of Figure 7 where the two obtained clusters are marked with blue and cyan nodes. On the left column of the figure, we can see that for the two clusters which neglect CA constraints, the cyan cluster spans across all CA's, while on the right the blue nodes correspond to the entire CA. An increase of the number of bidding zones leads to separation of the biggest outlier from the bottom bidding zone which is node 24 (middle row in the right column in Figure 7). Placing the outlier in a separate bidding zone diminishes the Ward's variance significantly enough that requesting a larger number of bidding zones influences other CAs but not the green one. In comparison, the left column in Figure 7 shows the results of clustering into an equivalent number of bidding zones but without implemented control area constraints. If the constraints are relaxed and the two bidding zones are required (top row), we end up with the bottom CA being split into two parts.

## 7. Summary

There are LMP and PTDF-based algorithms that can help in the process of bidding zones delimitation by taking into account economic and technical information about grid topology and generation units. These algorithms can be modified to address the governments' and TSOs' needs related to CAs consistencies. In this paper we showed that (i) an agglomerative algorithm based on LMPs can be adapted towards CA constraints, (ii) the outcomes with and without constraints become closer to each other if the number of bidding zones is increasing, (iii) the CA constraints decrease the computational and space complexities of the agglomerative algorithm, and (vi) including CA constraints leads to a greater non-monotonicity of dendrograms compared to including topological constraints only. Nevertheless, the latter effect is explained and does not influence the consistency of the result of the grid partitioning.

Additionally, as was already mentioned, the proposed adaptation of the clustering algorithm to the CA constraints is not contingent on the choice of LMPs as the clustering feature, and can be used as a general (or "meta") solution in approaches that take into account different characteristics of the grid model to perform bidding zones delimitation.

An intriguing area for future research is to apply the CA constraints algorithm on real-world data of an electrical grid-preferably on the model of the European continental grid-and compare the outcomes obtained from partitioning on different clustering features of grid nodes (LMPs, PTDFs etc.). Additionally, adaptation of the CA constraints algorithm for clustering based on features of grid interconnections (rather than nodes), such as congestion frequencies or dual variables on transmission constraints in Market Coupling solution, is another research task that is seemingly worth pursuing.

**Author Contributions:** Conceptualization, K.W. and M.B.; methodology, M.B. and M.J.; software, M.B.; validation, M.J., M.B. and K.W.; formal analysis, M.J. and M.B.; investigation, K.W. and M.J.; resources, K.W.; data curation, M.J.; writing—original draft preparation, M.B., K.W. and M.J.; writing—review and editing, M.J.; visualization, K.W.; supervision, M.B., K.W. and M.J.; project administration, K.W. All authors have read and agreed to the published version of the manuscript.

**Funding:** This research received no external funding.

**Data Availability Statement:** Not applicable.

**Conflicts of Interest:** The authors declare no conflict of interest.

## Abbreviations

The following abbreviations are used in this manuscript:

| | |
|---|---|
| MC | market coupling |
| CA | control area |
| BZ | biding zone |
| TSO | transmission system operator |
| LMP | locational marginal price |
| PTDF | power transfer distribution factor |
| OPF | optimal power flow |

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
