# Peer review of "Partitioning Power Grid for the Design of the Zonal Energy Market While Preserving Control Area Constraints"

_electronics, doi:10.3390/electronics10050610_

Round 1
Reviewer 1 Report
The authors study on Partitioning Power Grid for the Design of Zonal Energy Market is really good. I enjoyed reading the manuscript and the details of the study are clearly discussed. The article could attract large readership. I would suggest the article for publication without any revision.
Author Response
Thank you very much for the review, and positive marks. We updated the language, and we also corrected many elements in the text to make it easier to read and understand.
Reviewer 2 Report
MDPI: Energies |
Partitioning Power Grid for the Design of Zonal Energy Market Preserving Control Area Constrains |
Comment for the Authors |
The manuscript topic is interesting. The proposed methodology seems novel. Unfortunately, the authors should better explain the reasons behind this work and the critical issues in the literature.
The state of art needs improvements. I suggest redesigning section 1 and 2. It should be better if the authors follow the traditional manuscript workflow:
This should both help the reader’s understanding and enhance text fluency.
The English needs a proofreading. Several parts are not much clear. Algorithms and tables need a careful analysis.
In my opinion, the manuscript needs some clarification about the machine learning part.
The quality of some figure is very low. I suggest improving them for a quality research work. Some figure may be merged to enhance their descriptive power (Please see details below). The experimental results should be expanded (Please see details below).
Please, forgive me for any mistake made during this review. |
Questions: |
|
1. With reference to introduction, page 2 |
2. With reference to rows 79-80, page 3 |
3. With reference to Algorithm 1, page 5 |
4. With reference to rows 136 – 142. |
5. With reference to Figure 2. |
6. With reference to Table 1. |
7. With reference to the proposed methodology. Did you perform a preliminary analysis to choose the clustering algorithm? What are the results. Did you try other algorithms? Benchmarks are fundamentals in machine learning applications. |
8. With reference to Figure 3b. |
9. With reference to Algorithm 2. Did you check the output of your word/LaTeX processor before to upload the manuscript? |
10. With reference to rows 222-224 “Summing up, the problem of lack of monotonic growth of the dendrogram may provide an 223 important insight into the interpretation of relations between groups of buses and their connectivity as 224 described above, but it does not question the validity of hierarchical clustering methodology.” |
11. With reference to Algorithm 3. Check the output of your word processor. The readability is dramatically low. |
12. With reference to Algorithm 4. Check the output of your word processor. The readability is dramatically low. |
13. With reference to rows 117-118. Please describe better what are the features characterizing each object. It is not clear |
14. With reference to the proposed methodology: If I understood well, the chosen cluster number is not related to the chosen cutting level on the dendrogram, but it is constrained by the control area partitioning algorithm? Is right? It is not much clear. |
15. I suggest comparing the proposed methodology with a different zonal market area partitioning model. In my opinion, the employment of a traditional hierarchical clustering does not allow an exhaustive benchmark. |

Author Response
First of all, I would like to thank the reviewer for the effort spent on analyzing the text of the manuscript and the many comments which help us to improve the quality of the article. Below I provide responses to the comments:
Comment for the Authors
The manuscript topic is interesting. The proposed methodology seems novel. Unfortunately, the authors should better explain the reasons behind this work and the critical issues in the literature.
Response: In the introduction, we significantly extended the explanation of the problem. It is a novel problem that derives from the needs of recent changes in the European Union energy market design. There is an ongoing process, but there is a lack of tools that are applicable to solve it. This paper addresses these issues by defining and extending the approach which is known so far. Unfortunately, non of the known approaches were able to solve the control area constraints, and according to the best of our knowledge proposed solution is the first one which addresses these need.
The state of art needs improvements. I suggest redesigning section 1 and 2. It should be better if the authors follow the traditional manuscript workflow:
Introduction -> Critical Issues in the literature -> Limits of the previous works -> Proposed Contributions.
This should both help the reader’s understanding and enhance text fluency.
Response: We have tried to redesign the structure of the manuscript so that there is a better flow, and the reader could better follow our main idea.
The English needs a proofreading. Several parts are not much clear.
Response: We have corrected the text and tried to make id easier to understand. New parts are marked in green. Moreover many parts of the text were removed.
Algorithms and tables need a careful analysis.
In my opinion, the manuscript needs some clarification about the machine learning part.
As an example, why the authors chose Hierarchical Clustering and not a different technique? What could be the effect of a different clustering algorithm on the effectiveness of the proposed methodology?
Response: In fact, our algorithm extends the work of Burstedde who proposed to utilize LMP based descriptors of the nodes, and he also used the hierarchical clustering method. (this is directly indicated in the text - line 66). This method is the first choice because the most commonly used methods like k-means or other gradient-based clustering methods do not allow to include network structure as a constraint. On the other hand, the problem of network clustering is commonly called community detection, where many algorithms were developed, but these algorithms tend to identify weakly connected nodes which is not the case we try to solve. In our problem, we need to cluster the nodes described by some descriptors, here the LMP prices, and the network structure is an external constraint on which nodes can be grouped together. In other words, hierarchical clustering is the first choice algorithm able to solve it. Moreover, it has the nice property to visualize the structure of the data by the dendrogram, so it does not need to know the number of clusters in advance. Finally, its tree structure allowed us to implement the control area constraints.
[1] Burstedde, B. (2012). From Nodal to Zonal Pricing - A Bottom-Up Approach to the Second-Best. Proceedings of the 9th International Conference on the European Energy Market, 1-8.
The quality of some figure is very low. I suggest improving them for a quality research work. Some figure may be merged to enhance their descriptive power (Please see details below). The experimental results should be expanded (Please see details below).
Response: We have addressed these issues. None of the figures are vector graphics (EPS) so that they scale with the text.
Questions:
- With reference to the introduction, page 2
The critical issues in the current market coupling area partitioning models were not described. Please, improve this part in order to facilitate the reader in appreciating the potential benefits of the proposed methodology respect with to the state of art.
Response: We extended the text and more clearly defined the issue we try to solve. We also restructured the introduction and added a new section. Also, a more detailed explanation for the need of including the control area constraint is now given in the Introduction section.
- With reference to rows 79-80, page 3
Please, you should explain the reason for the inapplicability of the described methodology to your zonal market area partitioning problem.
Response: In the extended version of the manuscript, we focus on explaining this issue. As indicated above there is no such algorithm that addresses the control area constraints out of the box.
- With reference to Algorithm 1, page 5
Please, check the structure. The reading is difficult.
Response: The structure of Algorithm 1 has been improved.
- With reference to rows 136 – 142.
I suggest describing the algorithm in a more formal style. The small caps are absolutely necessary?
Response: The “small-caps” convention follows the default Latex algorithm style and is quite widely spread in the literature (for example, some major AI or algorithms textbooks, like Russel&Norvig “Artificial Intelligence, A Modern Approach”, or Cormen et al. “Introduction to Algorithms”). We find it quite readable, although we apologize that the first version of the manuscript had those sections difficult to read due to discrepancies in LaTeX styles and it appears only in MDPI class.
- With reference to Figure 2.
I suggest joining figure 2 and figure 3. Alternatively, you should put them as closest as possible.
Response: We left figure 2, but we also added the dendrogram to figure 3 to make it easier to compare the two dendrograms.
- With reference to Table 1.
Please, adjust the style of this table. The title position is wrong. Usually, the table titles are on top of the figures. Check this.
Response: The styling of this table was improved.
- With reference to the proposed methodology. Did you perform a preliminary analysis to choose the clustering algorithm? What are the results? Did you try other algorithms? Benchmarks are fundamentals in machine learning applications.
Response: We answered this question above, but shortly, there is no method that directly solves the problem defined by the EU to partition the energy market preserving control area constraints. Other attempts to clustering energy markets used standard algorithms mostly hierarchical clustering [1]. Moreover, hierarchical clustering is the most intuitive way to apply control area constraints, which is hierarchical by its nature. We added a short explanation to the text.
- With reference to Figure 3b.
Please add a legend. What the different colors (blue and red) stand for?
Response: We updated the caption of that figure, as well as extended the description in the text which refers to Figure 3,.
- With reference to Algorithm 2. Did you check the output of your word/LaTeX processor before uploading the manuscript?
What bold D stands for?
Response: The styling of Algorithm 2 was improved. “Bold D” stands for distance matrix - it was introduced already at the beginning of Section 4 (line 194) . We follow the notation where bold capital letters indicate matrix, bold letters indicate vectors and small normal letters indicate scalar variables.
- With reference to rows 222-224
“Summing up, the problem of lack of monotonic growth of the dendrogram may provide a 223 important insight into the interpretation of relations between groups of buses and their connectivity as 224 described above, but it does not question the validity of hierarchical clustering methodology.”
Please, How did you prove this statement?
Response: The problem of lack of monotonicity is popular in hierarchical clustering. For example, it appears when using centroid linkage without any constraints. Ward linkage is free of it, but the constraints, even the topological cause this effect. Still, from the mathematical perspective, the algorithm is represented as a binary tree, so it can be cut at any level producing the desired number of clusters. Only the dendrogram may look less intuitive, but it indicates that when merging two clusters the total variance of the market one is reduced in comparison to the variance of each individual. In other words, there were differences within each cluster in the nodal prices, but after the merge, the two price distribution compensates so that the total variance is reduced.
- With reference to Algorithm 3. Check the output of your word processor. The readability is dramatically low.
Response: The styling of Algorithm 3 was improved, it was caused by the MDPI Latex class which was in conflict with another package.
- With reference to Algorithm 4. Check the output of your word processor. The readability is dramatically low.
Response: the styling of Algorithm 4 was improved, it was caused by the MDPI Latex class which was in conflict with another package.
- With reference to rows 117-118. Please describe better what are the features characterizing each object. It is not clear
Response: The features of each node are the Locational Marginal Prices - either one-dimensional or a set of LMPs for each node. The LMP’s obtained for different network conditions - summer pick, winter pick, summer of pick, winter off pick etc. It is described explicitly in footnote 6.
- With reference to the proposed methodology: If I understood well, the chosen cluster number is not related to the chosen cutting level on the dendrogram, but it is constrained by the control area partitioning algorithm? Is right? It is not much clear.
Response: We see no constraints on the chosen number of clusters introduced by the CA constraints. There is an issue (due to non-monotonicity of the dendrogram after the introduction of a topology or CA constraints) of a less obvious relationship between the visualization of the cutting level of the dendrogram and the number of clusters, which is disputed in the text, but in general division into any number of clusters is possible.
- I suggest comparing the proposed methodology with a different zonal market area partitioning model. In my opinion, the employment of a traditional hierarchical clustering does not allow an exhaustive benchmark.
Response: As a benchmark, we have used the approach based on hierarchical clustering of LMPs with topological constraints (as used in Wawrzyniak et al., 2013, and derived from the Burstedde (2012) approach) to show in a most clear-cut manner the differences between results of only-topological-constraints and topological-and-CA-constraints methodologies. Moreover, according to the best of our knowledge, there is no other algorithm that satisfies CA constraints, so at the moment we want to show that the state-of-the-art solution can be updated to fulfill the CA constraints. The proposed algorithm can be also applied to the PTDF-based descriptors but we feel that it would yield a less readable outcome in terms of the influence of CA-constraints on the resulting bidding zones configuration.
Reviewer 3 Report
I have read your paper and there are some comments that must be improved.
1-In abstract you said "In order to address it, we develop a novel grid partitioning…" please don"t use WE. please modify to be anovel grid parioning is devlopoed or this paper develope….. also check this comment for the whole paper.
2- Key words must be alphabtically ordered.
3- langaue must be modified such as the first sentence in the introduction is not good formultion. The paper must be rewritten.
4- The algorithm written format is not good.
5- all figures must be modifed it is not clear although it has a big size.
6- In figure 6 you said that "Background color represent obtained zone delimitation…" my quest what is the difference between each color in this figure.
7- Summery section didn't clarify the findings of the paper and the future work.
8-The state-of-the-art description of the paper should be improved because the cited references are rather old ones. In the last years a lot of significant contributions had been made.
9-The theoretical background of the proposed method is not described adequately.
10- reference aren't in the required format style.
11- figure 5(b) title is clipped.
12-The authors have not fully explained what the novelty of this paper is - especially in the context of Electronics.
Author Response
First of all, I would like to thank the reviewer for the effort spent on analyzing the text of the manuscript and the many comments which help us to improve the quality of the article.. Below I provide responses to the comments:
I have read your paper and there are some comments that must be improved.
1-In abstract you said "In order to address it, we develop a novel grid partitioning…" please don"t use WE. please modify to be anovel grid parioning is devlopoed or this paper develope….. also check this comment for the whole paper.
The formulation in the abstract was improved along your suggestion.
2- Key words must be alphabtically ordered.
The keywords were re-arranged alphabetically.
3- langaue must be modified such as the first sentence in the introduction is not good formultion. The paper must be rewritten.
The language of the beginning of the Introduction, as well as of many parts of the whole text, has been reformulated.
4- The algorithm written format is not good.
We apologize that the first version of the manuscript had the algorithms difficult to read due to discrepancies in LaTeX styles and a conflict between package and MDPI style- it is now fixed.
5- all figures must be modifed it is not clear although it has a big size.
The figures have been re-generated to ensure proper quality.
6- In figure 6 you said that "Background color represent obtained zone delimitation…" my quest what is the difference between each color in this figure.
Figures were regenerated, and now we changed the background and node colors. Now the background color corresponds to the control areas and the node color represent obtained clusters. In our opinion that is more intuitive. Moreover, the colors are more contrastive..
7- Summery section didn't clarify the findings of the paper and the future work.
The Summary section has been expanded to highlight the advantages of the developed methodology as well as point out potential future work in this subject.
8-The state-of-the-art description of the paper should be improved because the cited references are rather old ones. In the last years a lot of significant contributions had been made.
Partially we updated the references, but most of the recent work in terms of network clustering was devoted to community detection (mostly motivated by the social network analysis), but relatively small progress is observed in energy market clustering especially in the context of the energy market managed by the Market Coupling.
9-The theoretical background of the proposed method is not described adequately.
The background has been improved and the description is corrected. All the updates are marked in green.
10- reference aren't in the required format style.
All of the references were moved to bibtex, and formatted according to the MDPI style
11- figure 5(b) title is clipped.
The figure’s title has been reformatted.
12-The authors have not fully explained what the novelty of this paper is - especially in the context of Electronics.
We extended the novelty of the article in the introduction. The paper addresses the problems related to Power systems, which is in the scope of Electronics and this particular special issue.
Round 2
Reviewer 2 Report
The Authors significantly improved their manuscript. They both clarified several requested aspects and revised the structure of the manuscript according my suggestions. The figure, and pseudocode quality is improved. Hence, there are not further questions from my point of view.
I hope the Authors will assess the proposed methodology to a real case study as proposed by them in the Conclusive section.
Author Response
Thank you very much for the review.
In the final version, we did small changes in the text related to the final spell check, performed by the native speaker. The language was standardized to British English.
We also did small changes in the bibliography, performing corrections related to conference materials.
Best regards
The authors
Reviewer 3 Report
the paper is modified. however, a bit modification to langauge is recommended. also reference is not in the required format.
Author Response

(The authors gave the same response as above.)
